# What Adolescents Have to Say about Problematic Internet Use: A Qualitative Study Based on Focus Groups

**DOI:** 10.3390/ijerph20217013

**Published:** 2023-11-02

**Authors:** Simone Rollo, Claudia Venuleo, Lucrezia Ferrante, Raffaele De Luca Picione

**Affiliations:** 1Department of Human and Social Studies, University of Salento, 73100 Lecce, Italy; simone.rollo1@unisalento.it (S.R.); lucrezia.ferrante@unisalento.it (L.F.); 2Department of Psychology, “Giustino Fortunato” University Benevento, 82100 Benevento, Italy; r.delucapicione@unifortunato.eu

**Keywords:** problematic internet use, adolescents, focus group, conceptualisation, intervention strategy

## Abstract

In this paper, the study presented is designed to gain a deeper insight into how adolescents describe, understand, and suggest dealing with Problematic Internet Use (PIU). Eight focus groups were activated with a total of 70 students from the 9th and 11th grades (Mean _Age_ = 15.53 ± 1.202; Female = 44.4%) in four different schools in Southern Italy. A Thematic Analysis was applied to the verbatim transcripts, and seven macro-categories were identified throughout the discourses collected: definition of PIU, symptomatology, impact, determinants, intervention strategy, opportunities and limits of the digital world, and needs that adolescents try to satisfy by surfing the net and which the offline world does not fulfill. Participants converge in seeing PIU in terms of addiction but adopt heterogeneous viewpoints in talking about the reasons for problematic engagement and possible preventive intervention strategies. In the overall picture emerging from the responses, PIU appeared to be the outcome of a psychological dynamic emerging from the interaction of individual, interpersonal, and sociocultural dimensions.

## 1. Introduction

In the twenty-first century, daily life without the internet is almost unthinkable for most people. The easy use of technology and online accessibility offer different opportunities in terms of communication, information seeking, social interaction, and connecting people around the world [1,2]. Especially for so-called “digital natives” [3,4], fully immersed in technology from their birth, the difference between “normal” and “problematic” Internet use is very fuzzy [5].

Here, the expression Problematic Internet Use (PIU) is adopted as an umbrella term to encompass any use of the internet that has a maladaptive impact on people’s daily lives, that is, preventing the subject from functioning fully in their life projects and social relationships, and that is accompanied by suffering and changes in their relationship with the world [6,7]. Relationship breakdowns, neglect of social life, and school or academic failure are a few examples of maladaptive correlates of internet use [8,9,10].

Although there is not a widely accepted label for internet-related behavioural problems [8], most of the terms used convey the idea of a disease—e.g., Internet Addiction [11,12]; Pathological Internet Use [13]; Compulsive Internet Use [14], and Virtual Addiction [15]—whose determinants, by definition, have to be sought in the individual with deficits of the brain (biological, neurological) and mind (cognitive or affective) that act as sources of origin: neurobiological factors [16,17], personality traits (e.g., impulsiveness, neuroticism, harm avoidance, reward dependence traits) [18,19], metacognitive beliefs [20,21] and mood disorders [22,23].

Consistently with this approach, prevention strategies are found to focus on vulnerable/at-risk individuals or age groups and their cognitive, emotional, and behavioural components: developing awareness of internet use and the effective use of time, increasing self-esteem and reducing anxiety, providing knowledge on different types of addiction, and common symptoms of dependency, with short- and long-term impacts being among the main areas addressed by the intervention (for a review: [24]).

In this paper, within the frame of a socio-constructionist perspective [25,26], the view of PIU as a disease is recognised as a narrative endeavour [27]: it is not limited to describing but establishes what counts as data [28], who is called upon to intervene (e.g., mental health professionals, doctors, family, schools, politicians), who has to be identified as the target of the intervention (individual, community), and on what aspects. Different models of explanation and conditions of observation end up defining different problems [29,30]. In the well-known rat park experiment, Alexander and colleagues [31] placed several rats in a cage and provided them with colourful balls, good rodent food, and tunnels in which to exercise. In such an environment, the rats seemed uninterested in taking cocaine-contaminated water, unlike rats placed in a cage without other rats to play with or any other stimulus. In choosing a mode of observation, we end up seeing some things and not others (we see the sick or maladaptive individual, not the context that nurtures or does not impose constraints on the distress).

Two main characteristics of the disease model are emphasised here.

First, the problem is located apart from the individual’s interpersonal network and social environment. However, if we think of adolescents—the most at-risk demographic for PIU—they are also children, students, and citizens; namely, they develop within an intricate and multi-layered social dynamic and specific family environments, specific school environments, and specific neighbourhoods that can offer resources or place constraints on their adaptive potential, from at least two points of view. On the one hand, this context may act as a source of malaise: for example, low family support, family functioning and parental monitoring, school climate, low social capital (e.g., inter-personal and social trust, size of social networks, social support), feelings of anomia, feelings of loneliness, and alienation were found to have a significant association with PIU [32,33,34,35,36], gambling [37,38], as well as other indicators of psychological distress [39,40,41,42,43]. On the other hand, the context may or may not offer resources and opportunities essential for young people’s growth in terms both of material and instrumental conditions (e.g., educational and professional opportunities; recreational settings such as cinema, theaters, and cultural associations) and semiotic resources underlying ways of perceiving and dealing with the problems in life. For instance, Ferrante and colleagues [44] found that the feeling of an adolescent expressing higher social malaise and a higher level of PIU is that nobody makes an effort to improve the present and the future of the country, not the ordinary people, not the politicians, not other institutions, so that ‘you can only live day by day’. Conversely, the feeling of congruence between the individual’s demands and the environment’s responses works as a protective factor for adolescents, preventing them from using the internet as a maladaptive compensation strategy. Similar findings have been made regarding other hazardous behaviours, such as drinking and gambling [45,46].

Second, within the disease model, the sufferer or risk group identified as the potential target of the intervention is considered a poor historian, unable to interpret the true nature of their condition [47,48], thus, their narratives tend to be considered of no use in understanding their subjective experience. On the opposite side, the constructivist approach invites us to conceive and value the sufferer or the individual at risk as a semiotic subject for whom “the historically acquired meaning of a situation or stimulus event is the major constraint on his or her response to it” [49] (p. 402). According to this view, meanings are interpretative categories able to guide and constrain people’s way of feeling, performing, and talking about their social world [50]. They organise the ways people think about what happens in their lives, live their lives, define what is problematic and what is not, make sense of their problems, evaluate whether or not to ask for professional or informal help, and relate to the therapeutic or preventive strategies addressed to them [48,51]. Two main points conveyed by this assumption must be underlined. Firstly, the acknowledgement of the performative value of meaning [52]. People’s ways of interpreting experience are not merely abstract judgments, they are a way of experiencing the material and social environment, of being channelled to act and react in a certain way. Secondly, the acknowledgment that meanings do not develop in a social vacuum. The “historically acquired meaning” of a situation has to be conceived as the result of field dynamics (sensemaking), where individuals situated in a system of activity (e.g., available resources and opportunities) and culture (e.g., social norms and values) recursively interact with each other [53,54,55]. Social processes (e.g., the media, scientists, health policies) as well as the here and now of the systems of activities where people experience their life (e.g., family, teachers, peers) influence how people make sense of their outer and inner realities and how people come to describe, explain, or otherwise account for the world (including themselves and their behaviours) [56]. In this sense, taking the meaning that people give to their experience as “data that counts” allows us to go beyond the contrast between the individual (subject, self, person) and the social (society, community), while questioning the very meaning of such a contrast.

On a methodological plane, recognising the mediating value of meaning on people’s ways of living, acting, and evaluating their experience orients the research towards methods of analysis that favour the exploration of people’s ways of thinking and relating to the problem under investigation (typically via focus groups and in-depth interviews). Narratives are important not because they furnish an accurate picture of what PIU actually is and how it should be conceptualised but because they are the means by which people understand and live their experience [57]. They also provide unique insight into the connections between individuals and society since people, by talking, participate actively in the practise of a particular culture. Narratives take place within the context of ongoing social debates and dominant narratives (widely accepted stories) about a phenomenon [58] and are shaped in part by a speaker’s awareness of and response to these, in terms of agreements, disagreements, and counter-arguments [59,60], sometimes challenging what is assumed to be true about the population under study.

On the plane of intervention, an in-depth exploration of what adolescents have to say about PIU through their narratives on PIU is a crucial step in planning effective preventive strategies. Indeed, if we recognise the mediating role of meaning on how people live their experience, no intervention strategy can proceed ‘in spite of’ the interpretative criteria of the interlocutors to whom it is addressed; we cannot take it for granted that adolescents agree with the “experts” or adults on what has to be considered “problematic” and on what has to be changed [61,62,63]. Just think how meaningless a discussion about the risks of the internet might seem to adolescents who consider risk-taking a way of shaping one’s family and social identity or to adolescents who—living in environments with no opportunities for personal fulfilment—cannot imagine their future and live only in the immediate present.

As mentioned, although the so-called “digital natives” are the largest risk group to whom prevention strategies on PIU are addressed, little research has been conducted to understand their perspectives. The current paper tries to bridge this gap. A qualitative study will be presented, designed to explore what adolescents have to say about PIU, whether they recognise it, how they explain PIU (its determinants), and which strategies for intervention they suggest.

## 2. Methods

### 2.1. Participants

This study was conducted in a middle-sized town in Southern Italy (Lecce-Apulia). Four high schools were asked to collaborate on this study (two in the urban area and two in the surrounding rural area). In each school, two focus groups were conducted, one with students from 9th grade and one with students from 11th grade.

Two students (one male and one female) were randomly selected from the subgroups of males and females in each 9th and 11th grade class of the single school. The selected students belonging to the same grade were then aggregated for the focus groups. In total, 8 focus groups, with 6–10 students, were conducted (Table 1), with a total of 70 students from the 9th and 11th grades (Mean _Age_ = 15.53 ± 1.202; Female = 44.4%).

### 2.2. Research Topics

An interview guide was defined as a list of questions [64], which directed conversation within each focus group towards three research topics: (1) When can we talk about PIU (i.e., what—in their view—constitutes a problematic use? How is it recognisable?); (2) How can PIU be explained? (i.e., what are—in their view—the causes of a problematic use?); (3) How can PIU be prevented? (i.e., what are—in their view—the target and the strategies of intervention?). Participants were encouraged to say whatever came to mind in response to these topics and respond in the manner that was deemed most appropriate, taking into account that the objective of the investigation was to collect their subjective view of PIU. The focus group leaders took care to foster an open conversation, allowing each participant to explore each of the three topics in a way that was meaningful to them and also allowing divergent points of view to be expressed.

Each focus group lasted 90 min, and the discussions were conducted by two psychologists in a private part of the school where they could not be overheard by teachers or peers.

In accordance with the Italian ethical code concerning the protection of personal data (Legislative Decree No. 196/2003), participants were informed about the general aim of the research, the anonymity of responses, and that the discussion would be audio-recorded. It was verified that each student had their parents’ permission and informed consent signed. No incentive was given. All procedures were approved by the Ethics Commission for Research in Psychology of the Department of Human and Social Sciences of the University of Salento (Lecce, Italy) (25 March 2021; protocol no. 0056300).

### 2.3. Data Analysis

It is worth noting that the discourses of adolescents adopted a connectionist, rather than disjunctive, logic in approaching the topics proposed: the statement of one was commented on, deepened, or supplemented by another, so that the overall discourse emerging from the focus groups held together viewpoints, subjective ways of feeling, and direct and indirect experiences. For this reason, when analysing the transcripts qualitatively, we chose to identify and code the proposed themes, but it did not seem possible or appropriate to analyse them in terms of frequencies.

Focus group interaction was transcribed verbatim and read as many times as was necessary to obtain a sense of the essential features without feeling pressured to move forward analytically [65]. Then, Thematic Analysis (TA) was applied in order to identify and systematically organise the students’ narratives [66] into patterns of meanings (i.e., themes). We began the analytical work by listing the themes the interviewees talked about within each of the questions proposed and labelling them: for example, for the topic “when we can talk about PIU,” utterances focusing on the time spent on the internet were grouped under the label “high frequency of internet use,” statements focusing on the compulsive desire to get connected were collected under the label “craving,” and so on. We then grouped themes into macro-categories (e.g., “high frequency of internet use,” “craving,” and “interpersonal conflicts” were grouped under the macro-category “symptomatology”).

TA was performed by two groups of three researchers (a total of six research collaborators) that worked independently. Each theme was consolidated after an intra-group (three-researchers) discussion and subsequently through an inter-group (six-researchers) discussion. Thanks to these comparisons, common judgments and differences among researchers emerged in the formulation of a synthetic label. The common judgement led to the selection of the theme identified; any divergence was resolved by referring to the scientific literature (e.g., labels referring to the compulsive desire to get connected were redefined in terms of “craving”) or to the more general discussion, thus seeing how theme X is connected within a sequence of questions and answers (e.g., the label “social detachment” was used to group both utterances referring to symptoms and utterances referring to the impact of PIU, because participants referred to social detachment both in the attempt to define the symptoms of PIU and its consequences). When the disagreement could not be resolved between the researchers, an external researcher (research coordinator) was involved as supervisor.

## 3. Results

TA allowed us to identify seven macro-categories. Five relate to PIU: (a) Definition; (b) Symptomatology; (c) Impact; (d) Determinants; (e) Strategies of Intervention. Two further topics focus on the common conditions of living in a digital world: (f) opportunities and limits of the digital world; and (g) needs adolescents try to satisfy by surfing the net, which the offline world does not fulfil. In the following, for each macro-category, specific themes and fragments of discourse are reported.

It is worth noting that, in terms of definitions, symptomatology, and impact, adolescents in our focus groups largely agree with a view of PIU as addiction. However, when asked about the determinants of PIU and possible intervention strategies, their discourses become more heterogeneous, and the role of the interpersonal, social, and cultural environment in the construction as well as in the resolution of PIU is largely emphasised. Together and through these aspects, participants solicit us to understand the meaning of internet use, a meaning that can be captured in the light of the socio-historical context in which they live and the developmental challenges they face and try to solve through social networks.

(a)Definition of PIU

One single theme is recognisable in the discourses of participants when they define PIU (Table 2). Across focus groups, participants embrace a perspective based on models of substance-related addictions (e.g., tobacco, alcohol…), pathological gambling, or Internet Gaming Disorder—which is also the most widespread way to describe and conceptualise PIU by researchers and health professionals [10,67,68], for review: [69].

Addiction is often defined by the participants as a “vicious circle,” where the negative effects of being connected (e.g., isolation) work as triggers with respect to the need to connect again and find a refuge from problems in the virtual world.


*Basically, it is like a vicious circle because you lose self-esteem, you take refuge in the internet and then even more… then, one does not leave the house, does not train, does not eat, loses friends who are the support network and then it’s a vicious circle.*


(b)Symptomatology

Consistent with the dominant view of PIU as addiction, a (b.1) high frequency of internet use, (b.2) abstinence, (b.3) impaired control over the activity, (b.4) craving, (b.5) social detachment, and (b.6) interpersonal conflicts are the symptoms mentioned by participants. A further symptom and manifestation of PIU is recognised in enacting (b.7) socially inadequate/unacceptable behaviours (e.g., violent actions learned online or posted on a Social Networking Site, sharing of hot/provocative photos, or linking to sites prohibited to minors). An overlap between what is illegal and/or morally inappropriate and what is problematic can thus be observed (Table 3).

(c)Impact of PIU

The participants in our focus groups show they have a good awareness of the risks associated with PIU, as if to offer a kind of response and counterargument [59] to the vision narrated by scholars, families, and teachers, where adolescents tend to be perceived and described as an unaware group with little ability to judge risk. A copious list of aspects—ranging from psycho-physical malaise to impairment of social functioning and the ability to interact adaptively within contexts—emerges from their comments: (c.1) detachment from reality (“it makes you believe things that don’t really exist”), (c.2) mood alterations (e.g., anxiety, depression, sense of loss, emptiness), (c.3) physical problems (e.g., stinging eyes, fatigue, sleep reduction, loss of concentration), (c.4) loss of critical thinking and exposure to risks (e.g., cyberbullying, self-harm…), (c.5) personal negligence and poor school performance, (c.6) social detachment (no longer being willing and able to communicate with the outside world, apathy, withdraw into oneself) (Table 4).

(d)Determinants

With respect to determining factors, the participants’ discourses prove to be heterogeneous. Three different models of understanding PIU coexist in their discourses (Table 5).

One is consistent with the view of PIU as an “addiction”. We place here the participants’ utterances referring to (d.1) the structural characteristic of internet devices (rewards and other features of online games that encourage one to stay connected for many hours) depicted as having the power to keep people constantly online, (d.2) individual determinants (personality traits, emotional states such as sadness and boredom) that would prompt the use of the internet as a means of emotional regulation and (d.3) age-group determinants (i.e., adolescents’ vulnerability: their tendency to feel insecure, to feel they are not accepted, but also their ‘innate’ condition of being born with a smartphone in their hand which makes them predisposed to addiction). In spite of the diversity of the determinants cited, it is generally agreed that PIU is due to an inherent vulnerability (whether or not it is encouraged by the medium) over which the individual has no control.

Another model of understanding PIU is consistent with a “relational perspective” and focuses on the role of parents and friends. Within this cluster of discourses, attention shifts from within individuals to what happens around them. Different interpersonal factors are evoked, such as (d.4) low parental monitoring and bad parental example, (d.5) poor presence and quality of parental attention, (d.6) isolation, and lack of integration into the peer group. The participants’ discourse suggests that when families appear to be unsupportive (distracted, uninterested in the individual’s problems and projects) and/or the peer groups appear to be rejecting or non-inclusive, the internet can offer a larger, virtual community where one can find answers to one’s needs for recognition and belonging, a way of escaping from life’s problems, a relief from emotional-affective distress.

Finally, a view of PIU consistent with a “socio-cultural perspective” is proposed: emphasis is put on the idea that individuals’ patterns of thought and action are nurtured/constrained by the structural conditions and cultural environment adolescents inhabit. The focus is placed on (d.7) the lack of alternative channels for socialising, having fun, and spending time, as well as (d.8) the role of influencers and related cultural models that suggest that money and popularity are the only way to live life.

It is worth noting that in the discourses of the participants, the three explanatory models are not mutually exclusive. For instance, in the following fragment, the intertwining of the interpersonal and social spheres is emphasised: permeability to socially proposed criteria of success and self-affirmation is framed in the situation of not having alternative guidance and reference-points.


*Nobody tells us what to do, so when someone tells us what to do (in reference to social groups and influencers) we feel safer listening to their advice.*


(e)Strategies of Intervention

The three explanatory models mentioned above convey a different view of what kind of problem should be addressed by the prevention strategies and of what the target of the intervention should be (Table 6).

Consistently with the focus on the individual, (e.1) the professional help of a psychologist at a treatment centre is identified as the means to counter individual vulnerability, for instance, by improving self-esteem or, more broadly, self-image, or by doing a digital detoxification.

Consistently with the focus on the relational determinants, emphasis is placed on the preventive role of the interpersonal environment: (e.2) more parental monitoring of children’s internet use; (e.3) appropriate parental educational styles and family climate; (e.4) friends’ support; (e.5) education activities to promote informed use of the internet. The family should suggest alternative channels of leisure and entertainment and stimulate interest in activities to perform in the ‘real’ world; they should play a regulatory role, imposing and maintaining boundaries and limits in the use of the network (e.g., limiting access to the internet, restricting sites forbidden to children or adolescents…), they should communicate interest in and care for what the child does. The network of friends should recognise the signs of discomfort, listen, offer support, and say what is wrong and what is not. The importance of education in the use of the internet, the opportunities it offers, and the risks and dangers it exposes are also underlined. Participants point out that not all adolescents are supported by parents or teachers who can guide them in responsible ways. Education for a responsible use of the network should be of interest not only to adolescents but also to their main reference systems, i.e., families and schools.

Consistently with the focus on the social determinants, the role of “the social policies” is pointed out: (e.6) opportunities for socialisation and leisure; (e.7) policies to limit access and ban inappropriate sites; and (e.8) promotion of more healthy identification models. These adolescents, often described as individuals without control, require more restrictive policies and more vigilant supervision from institutions, underscoring the importance of working to prevent minors from accessing certain sites or making some online games subject to parental consent. The importance of promoting healthier values and identification models (through advertising campaigns, cartoons, and films) is also underlined, to emphasise again that problematic behaviours on the internet are fuelled by inappropriate messages and cultural models in wider society, as well as by a need to belong and be seen and recognised that does not always find satisfaction in the offline world.

(f)Opportunities and limits of the digital world

While capable of identifying problems related to excessive use of the net, teenagers also invite us to recognise how, in the twenty-first century, daily life without the internet is almost unthinkable for most people. The state of being digital natives and living in a world dominated by technology is not idealised but simply recognised as a new, different condition that has changed ways of spending free time, playing, and meeting friends.


*I don’t think there is a single person who can manage without a smartphone. Thanks to the smartphone we can socialise. If you call your friend, you can meet him. How can you survive without a smartphone? Without your smartphone, you are locked up at home.*



*Before, for example, when the smartphone was not in use, people and the world were more active. For example, to go out we went to a friend’s house. There was more interest in other things. My father always tells me that he went out with friends, they messed around… they jumped the walls in the countryside to pick prickly pears.*



*Technology did not exist before, not even appliances. Everything was done by hand, there was no vacuum cleaner… you made do with what you had… Then teenagers played in the street with stones.*



*Although we are young, the difference between when we were children and the child of today is very noticeable… sometimes even in the restaurant you see children who sit still watching a screen for the whole meal… that is, it is not something that we did when we were kids. As children we used to take dolls to restaurants or toys. I (played with) toy cars. I used to bring crayons to colour in…*


In their opinion, something has been gained, but something else has been lost. The easy use of technology and online accessibility offer different opportunities in terms of: (f.1) performing daily tasks more easily and/or learning more quickly; (f.2) spending time; (f.3) information seeking; (f.4) sharing ideas or interests, and approaching people. Alongside the opportunities, various limits or risks are also recognised on the internet, even when there is no question of problematic involvement: (f.5) meeting people that are not what they seem; (f.6) being exposed to untrustworthy/dangerous people or applications; (f.7) being exposed to untrustworthy news; and (f.8) being exposed to a false view of life (Table 7).

(g)Needs that adolescents try to satisfy by surfing the net and which the offline world does not fulfil

The participants confide—in this part of the discourse—their frailties; their insecurities; but also their strong need to dialogue; to meet people with whom they can identify and exchange ideas—something that does not speak of a mental disorder but of a demand for identity and recognition that is not always satisfied in the offline world. The internet serves the purpose of relating to (g.1) people you would not know how to approach in the off-line world, (g.2) finding what you do not find in reality (e.g., dialogue, feeling of belonging), and (g.3) nourishing self-esteem and self-image (Table 8).

## 4. Discussion

Our study aimed to explore “whether” adolescents recognise PIU, “how” they explain PIU (its determinants), and “which” strategies for intervention they suggest. Touching on these issues, the adolescents participating in the focus groups also recounted opportunities offered by the internet and questions of identity and recognition that they try to solve through social networks, even when they cannot describe this use as problematic. In other words, they tell us the meaning of their involvement and signal the importance of understanding it—an approach to internet use that is only marginally present in the literature [5,70,71].

The following offers a comment on the main findings emerging from the analysis of the transcripts; short summaries will be provided of the core messages that focus group participants seem to be delivering in response to the stimulus questions proposed.

### 4.1. We Know That PIU Can Be Seen as an Addiction…

Adolescents’ way of defining PIU does not differ from that widely shared by researchers, health professionals, and the media. PIU is defined in terms of an addiction, juxtaposed with other addictions in terms of manifestations and symptoms (i.e., high frequency of use, abstinence, impairment of control over the activity, craving, social detachment, and interpersonal conflicts). This is not surprising. The view of PIU as addiction is dominant among scholars and health practitioners, who offer specialised vocabularies to describe people’s experiences [72]. From a complementary point of view, Gergen argued that psychopathology categories are “socially connoted scripts placed within the sphere of social discourse, with which some individuals identify” [57] (p. 268). People can choose to describe a problem in terms of addiction (or a bipolar disorder, a social phobia…), not because this interpretation best fits the observable facts but because it is a view that serves useful purposes for themselves and/or for society in general [13,73]. For instance, with respect to gambling, several qualitative studies have shown that a view of gambling in terms of addiction serves the purpose of counteracting a “moral” representation of oneself as irresponsible and greedy [52,74]. In the case of the adolescents participating in our focus groups, the desire to show themselves as being conscientious and aware emerges not only in the use of specialised language to define PIU but also through their dwelling on various problems that they recognise as related to a misuse of the internet (i.e., detachment from reality, mood alteration, physical problems, loss of critical thinking and exposure to risk, personal negligence and poor school performance, social detachment).

### 4.2. Our Actions Take Place in a Context

Despite the reference to a view of PIU in terms of addiction, the determinants cited by the participants in their attempt to explain the onset of PIU are not confined to focusing on the “addictive character” of the medium and/or of its applications or on the individual characteristics that make individuals more likely to develop an addiction. Alongside these aspects, the quality of their relational network (e.g., parental educational styles, behavioural patterns proposed by adults) and their living environments (e.g., socialisation opportunities and leisure activities) is emphasised. Internet use and misuse appear to be strongly intertwined with what happens around adolescents and with material and immaterial resources made available by their social-cultural environment to face unpleasant feelings and deal with problems and developmental challenges. “We live in a context, and we cannot avoid modulating our behaviour accordingly, this is the message that they seem to deliver to adults and experts.

PIU emerges from the interaction of individual, interpersonal, and sociocultural dimensions.

Each of the explanatory models (addiction, interpersonal, and socio-cultural explanatory models) proposed by the adolescents who took part in the focus groups finds anchors in the scientific literature, where, alongside a prevailing focus on individual determinants, in the last twenty years models more attentive to the interplay between the individual and the interpersonal and social spheres have been proposed. For instance, compensatory models have suggested that people can gain emotional relief and fulfil their need for social contacts through internet devices [16,75]. Socio-cultural models argue that a model of PIU must focus on a player’s interpretations and evaluations of the meaning of internet use within a social and cultural context. They argue that more efforts should be made to better understand the role of social norms in shaping social identity and values and in predicting actual behaviour in virtual communities of the so-called “groups at risk” [46,76,77]. On first reading, we could therefore conclude that the adolescents interviewed do not add anything to what quantitative research has already shown.

However, for the frame that organises the present study, the value of the narratives collected derives not from their mere ability to represent “reality” or to explain the “nature” of Problematic Internet Use, but from their being the means by which adolescents understand PIU and the meaning of internet use and misuse. Their discourses inform us of how participants position themselves within the context of ongoing social debates and dominant narratives. On this point, it is worth noting that the discourses of adolescents—differently from what is usually the case among scholars—adopt a connectionist; rather than disjunctive; logic in identifying determinants of PIU. In the focus groups, there was no friction; the statement of one was commented on, deepened, or supplemented by another in the search for an explanatory framework that held together viewpoints, subjective ways of feeling, and direct and indirect experiences. In the overall picture emerging from the narratives, PIU appeared to be the outcome of a psychological dynamic emerging from the interaction of individual, interpersonal, and sociocultural dimensions.

### 4.3. Preventing PIU Requires Paying Attention to Multiple Spheres

Asked what can be conducted to counter or prevent PIU, participants recognise psychological help as the elective intervention for those manifesting PIU but emphasise how prevention requires a wider gaze to encompass the adolescent’s interpersonal and social environment. In focusing attention on the sick or maladaptive individual, adults and experts end up not seeing the context that nurtures or does not impose constraints on psychological distress. Prevention is a matter of quality in family and peer relationships, but it is also a matter of socialisation opportunities and leisure activities made available by the social environment. Countering addiction, from this perspective, primarily means working to increase the channels and opportunities to connect with others.

### 4.4. Consider the Meaning of Internet Use and Misuse

The value of understanding the meaning of internet use in a contemporary scenario is another point emphasised by the adolescents’ discourses. Participants in the focus groups recounted the opportunities offered by using the internet, recognised as an indispensable component of life, to facilitate communication, the immediacy of relationships, and daily activities. Even being disconnected and not having devices to surf the net can lead to functional impairment (i.e., a significant deleterious impact on daily life) in the contemporary world, where everything is based on online connections [78]. Alongside this consideration is the observation that—even when no symptoms of an internet addiction are recognisable—there is an important question of meaning, belonging, and identity that drives the search for an online connection. When—as scholars—we are engaged in identifying a precise boundary and cut-offs to establish the problematic nature of an activity or behaviour (as is the case in the categorical approach to psychopathology), we risk losing sight of the malaise and the painful subjectivity of adolescents who do not fall within our scientific definitions of at-risk or pathological groups.

## 5. Concluding Remarks

Many studies on PIU have focused on adolescents, recognised as the group most at risk, to whom prevention interventions should be directed. Nevertheless, little research has been conducted to understand the perspectives of the so-called “digital natives.”

The current study tried to bridge this gap with the idea that no preventive strategy can be effective without an understanding of the meaning that adolescents give to the use, even problematic, of the internet.

The participants in our focus group do not contest the dominant view of PIU as addiction, nor do they overlook the risks and negative impact that internet misuse can have on one’s life. At the same time, they suggest that knowledge of these risks is not always enough to prevent PIU because issues related to self-image, perceived quality of friendship and family relationships, lack of alternative and meaningful activities, and models and criteria for success proposed in the broader social environment are at stake.

Significant implications for policy can be recognised if this perspective is taken into consideration. First, any intervention that is limited to the specific domain of PIU is likely to have limited efficacy, given that adolescents (and more generally, people) shape their way of using the internet not only according to internet domain-specific beliefs and expectations but also according to their ways of representing themselves in relation to their significant others and in relation to the wider socio-cultural environment [46,79,80,81]. Preventing PIU is therefore not simply a matter of controlling access to the internet but of offering spaces for listening and reflection on the ways in which adolescents try to respond to their own needs for recognition and sociality. Second, if these ways are encouraged by the social environment, strategies should be sensitive to how adolescents’ network of interdependencies (e.g., family, friends, teachers) may frame and influence the ways they think and act and include this network in the range of action. The interpersonal environment plays an important part in seeing or not seeing individual or age vulnerability, in whether or not support is provided for the difficulties of growing up or, more broadly, of life, in whether or not healthy models of identification are offered, in whether limits and rules are set, and in whether or not alternative ways of spending time, having fun, and socialising are proposed. Preventing PIU, thus, means rethinking—sometimes radically—material, relational, and symbolic resources (e.g., socialisation channels, relationship models, educational styles, modes of communication, criteria of social recognition) that the environment makes available to interpret their experience, face problems, and make the future thinkable. The problematic nature of some patterns of internet use emerges in the interaction between the person and his/her worlds; thus, each of these levels needs each other and is dynamically, dialectically, and jointly made up by each other. This collective process appears to be underestimated in the current understanding of PIU.

### Limitations and Future Direction of Research

Some methodological limitations of our study should be considered. First, since it is based on a convenience sample, the results cannot be generalised. Characteristics such as the socio-economic status, educational background, and cultural background of the participants can play an important role with respect to what teenagers have to say about PIU. Second, the qualitative analysis of how adolescents represent and explain PIU could be improved by considering quantitative measures accounting for their internet usage patterns. As a matter of fact, we do not know how the participants in our research characterise themselves in this respect, and we cannot therefore exclude the possibility that the discourses collected refer to adolescents who make a balanced and adaptive use of the internet. Future research should consider other factors such as psychological well-being, parental monitoring, perceived social support, and a sense of belonging to the community to look more closely at the way individuals, their system of activity, and the socio-cultural scenario interact with each other in constructing the ways adolescents represent and use the internet.

## Figures and Tables

**Table 1 ijerph-20-07013-t001:** Group composition.

Variable (School)		Focus Group	Mean Age (DS)	N. Participants
9th	Urban	1	14.40 (0.548)	6
		2	14.56 (0.527)	9
	Rural	3	14.22 (0.441)	9
		4	14.25 (0.463)	8
11th	Urban	5	16.60 (0.843)	10
		6	16.60 (0.516)	10
	Rural	7	16.44 (0.527)	9
		8	16.33 (0.500)	9

**Table 2 ijerph-20-07013-t002:** Definition of PIU.

Theme	Fragments of Discourses
Models of substance-related addictions	Everyone thinks about smoking, drinking, drug addictions, but the smartphone is no exception! In my opinion, the problematic use is also due to the fact that, like drugs and gambling, the Internet can lead to pathological addiction… It’s like a cigarette, like a person who has been smoking for many years… […] first they started slowly [to use the internet], then they increased, so they can no longer do without it… now that is their dose, so… enough… like drugs. It is an addiction combined with technological tools!

**Table 3 ijerph-20-07013-t003:** Symptomatology.

Themes	Fragments of Discourses
(b.1) High frequency of internet use	[It is problematic] if you spend more than five- or six-hours using PC, PlayStation and so on… When a person uses their smartphones a lot, beyond two hours a day, that’s problematic use! A person who uses the internet in a problematic way is someone who is always connected… they don’t even look at themselves anymore… you see people who are also very unkempt who are always on the smartphone.
(b.2) Abstinence	Like someone who uses drugs, if they don’t use them for a while, they feel abstinence… also someone who uses the internet, if they don’t use it for a while, they feel abstinence too. For example, when you don’t use it [the internet] for a while, you feel anxious. You feel incomplete. In other words, you can be said to be in abstinence from lack [of the internet].
(b.3) Impaired control over the activity	So, if you receive a message, you cannot “not reply” that is, you are really tempted to see and reply! Can’t disconnect when it’s time to disconnect. They can’t stop anymore.
(b.4) Craving	Maybe, when you leave the house and you are happy to come home to play, like what happens with adolescents… I’ll give this example because it is more… maybe, when you smoke, you become addicted and obviously you can’t smoke in class and you always think “the break, the break, I have to smoke a cigarette!!”, the same with the smartphone… you can’t wait because you think that you must go immediately on your smartphone. I am talking to you… but I am with my thoughts on the smartphone… I have to open Instagram, I have already seen everything, but after five seconds I am still there, because it is normal, even if there is nothing to see, because I quit the App five seconds ago. My thought is that “I must necessarily open Instagram”.
(b.5) Social detachment	Maybe… someone doesn’t talk to anyone, they are always alone. For example, in the time that they are at school, for example at break, they don’t talk to their classmates, but they play on the smartphone. They don’t relate to others. They never talk, or in any case, when the teacher asks them something even then they have difficulty speaking… the person who is at the computer for many hours. However, while playing they have a bit of a problem talking, even with an adult, not just with their peers. The lack of socialisation… that is, a person recognises that they are a problematic user because it is tiring for them to socialise with others… because they are only interested in the smartphone or any other technological tool… and so also a little “apathy” towards others… For example, if you ask someone who is using internet something, they don’t even answer you because they are always on the smartphone. They don’t even take any notice of you because they are really into… they just can’t disconnect and say “I’ll pause the game…”.
(b.6) Interpersonal conflict	Impatience with others… If you take away the smartphone or tell them “use the smartphone less”, they get really angry, they react badly. There are people that smash the screen! When my mother takes it away… she [sister] starts screaming, really screaming “Why are you taking my smartphone away?”. [Without my smartphone] I was nervous… I could not break away, I was about to tussle to my mother… my cousin, for example, struggled with my aunt for the same problem.
(b.7) Socially inadequate/unacceptable behaviours	Sending inappropriate photos online… with those photos you can make fun of people or otherwise mistreat them online… so it’s also a symptom. However, in my opinion, a 10-year-old child who is on a Social Networking Site against their parents’ wishes… they [children] don’t realise what they are doing and where they can get to! Visiting inappropriate sites prohibited for minors under the age of 18. It’s inappropriate for us to use them!

**Table 4 ijerph-20-07013-t004:** Impact of PIU.

Themes	Fragments of Discourses
(c.1) Detachment from reality	The internet can make you see reality in a “distorted” way; therefore, it makes you believe things that don’t really exist. Consider the virtual as the real, that is, you enter a world that is not reality but automatically everything is reality. Internet leads you to reject reality, as things really are and say, “I prefer to be on the smartphone where I can be someone else!”. Dissociation, detachment from reality…
(c.2) Mood alterations	Then, it is also a psychological fear… if you take away their smartphone, the person suffers from anxiety or stress… There are people who feel completely lost without connection, they don’t know what to do, they can’t do anything without their smartphone. It causes nervous jerks… when you disconnect from the online world and someone tells you something you don’t care about, you get jittery jerks… Feeling emptiness inside your stomach too that completely blocks you, which makes you feel sick. Anxiety, panic, stomach pain and all. Take refuge in a room and think and stay with social media and see what you can do and not know what to do. Wanting to do something but failing. I think sadness… anxiety, desperation and nostalgia… because, sometimes it happens to me that when I find some Post a little sadder, I miss something, then there’s a tendency to self-harm and that’s it.
(c.3) Physical problems	In my opinion, some symptoms that the internet can cause are stinging eyes… because someone who stays in front of a monitor or screen for a long time or in front of something like a TV or a tablet or a smartphone, can find their eyes stinging… Because, in any case, spending many hours staring at a screen you feel tired and even if you rest, this tiredness remains over you. When the person sleeps, let’s say they have kind of spasms…For example, if this person is supposed to sleep at least eight hours a day, they say “I’m playing the last game” instead of sleeping. In the end, they only sleep five- or four-hours.
(c.4) Loss of critical thinking and exposure to risk	(Problematic user) always shares their personal information online… for example by talking to a person, even on Instagram Direct, for example. Direct is the chat. For example, you talk to a person and, starting to get to know you, they ask you for certain personal information… and you provide it because you trust this person. But then the person turns out to be a hacker who uses a fake profile to steal data from people. I think of the suicides, like cyberbullying, or other suicides like girls that commit suicide, “let’s meet in this place” and then maybe it ends badly too… because first they make you think positive things… There have been people who have even committed suicide or gone as far as killing other people because they have been banned from using (Apps and devices).
(c.5) Personal negligence and poor school performance	There are certain people you see really fixated with it, they do badly in school and they don’t play sports because they really prefer to be online, not really the same amount as us, but in continuation. Their life, like drinking, eating… depends on the internet! Sometimes some people don’t eat because they are connected. They just don’t feel the need to eat because they are too busy with games. There can also be non-commitment at school, and therefore low school achievement. Maybe, even the physical appearance. A neglected look. Practically, rather than thinking about eating and drinking, a person continues to be on the internet…
(c.6) Social detachment	For me the person becomes an asocial guy who can’t socialise… social detachment! But in the end, always being online leads you to close in on yourself. If you don’t socialise with others, you also lose some feelings towards people and therefore you become apathetic… because you live in a world of your own… It leads to non-communication with the outside world, to losing friends, to withdrawing into oneself and finding only that world.

**Table 5 ijerph-20-07013-t005:** Determinants.

Themes	Fragments of Discourses
Addiction explanatory model	
(d.1) Structural characteristics of internet devices	You cannot disconnect a game otherwise you have to start all over again. The problem is that new video games have games that if you turn them off while playing you get penalties; therefore, it forces you to finish the game and sometimes it even lasts hours. There are these games that really entice you to buy things. […] So [the type of game] makes you want to buy more.
(d.2) Individual determinants (personality traits, emotional states)	If a person is introvert and unable to relate face-to-face with others, by using the internet they can find a way. If a person is shy… if you are outgoing with people, you have no problem immediately showing what you are really like. It depends on the character! When you are feeling down, because it happens in adolescence, there are posts on the internet, pages that put up these self-defeating posts and you get even more depressed. You just want to read them, because you are in that mood, and they involve you in an absurd way. The posts, the songs are exactly coherent with the period you are going through… for example we are sad, we read those posts and they have even more effect on us, we tend to collapse. Me too, though, when it is a period when you are sadder and you want to be alone, you go onto Social Networking Sites and see some posts and you feel even sadder. Last night I was listening to music, I felt depressed… so I went to look for the profile of a guy […] who makes poems online, I read them… I don’t know, but reading his poems I became stable again, I was no longer sad.
(d.3) Age-group determinants	In my opinion, it is only in the period of adolescence, that maybe you are less secure, and you look for the internet, but in adults it is not. So, is it a question of safety, of feeling insecure? But not only with friends, it can also be with family, that is, if you do not feel accepted by your family and you shut yourself up on the internet looking for… Nowadays, children are already born with a smartphone in their hand. It is something that depends on their parents’ attitude, but it is innate.
Relational explanatory model	
(d.4) Low parental monitoring and bad parental example	[…] then people wonder why the child behaves like this [imitating the behaviour of the videos], obviously if the parents let them use the phone 24 h a day, without supervision, this behaviour is normal. A cause may also be when from an early age one begins to use telephones, computers, to go on the internet with no limits; therefore, we also say parents’ poor ability to bring up their children. Maybe they unwittingly allow them to do things from an early age just to please them, not knowing that the child can be easily influenced… Parents set a bad example when they don’t pay any attention to us being on the smartphone!
(d.5) Poor presence and quality of parental attention	For example, parents who are not very present in their child’s life may perhaps lead to the kids being addicted to internet. For example, the family who is not interested (in you) and then confides/vents on social networks. For example, the family ignores your interests, does not encourage you and so you are forced to let it go…
(d.6) Isolation and lack of integration in the peer group	If somebody isolates themselves from the group of friends, it means that they don’t get along with them, the latter isolate the person and so they play on the smartphone. Maybe… they find solace there by being online. Maybe… they just don’t know how to relate to others and so they go online, it’s not necessarily the others that have to isolate them. Maybe… they are in a group of friends where there is the bully who teases them and then they isolate themselves because they are being teased… Relationships with the people around you, I mean, if they don’t accept you, if there’s no community that accepts you, you are looking for someone who can accept you on the internet. But I meant that the causes are always the same, that is, being removed from a group, in this case of real people, and then going on the internet to look for others…
Socio-cultural explanatory model	
(d.7) Lack of alternative activities	If you have nothing to do, then it is natural that you choose to play a video game. I spend a lot of time on the phone because in the afternoon I have nothing to do… I watch TV Series… I am on Instagram… We use the smartphone especially when we have nothing to do to pass the time.
(d.8) Cultural models	To have more followers… Now we are almost all dependent on followers, on Instagram. Even to be a little famous, because we see influencers, so it would be nice to have a life like them and—in the end—one tries! We have been so influenced by the internet, by followers! Instagram is part of real life with restaurants, bars, wherever there is a connection… Now the whole world is connected. I am referring to those users who reach tens of millions of followers because you have a host of people, but most of the time they are kids with the same attitudes as you, because in the end they have the same thoughts as you: they all think the same way, they all dress the same, use slang phrases used by the same person… Now, for example, the fashion for teenagers is those who dress with brands like Gucci, Luis Vuitton get a lot of visibility and are followed on Social Networking Sites because they post photos only of clothing and become famous. (With reference to the influencers) I am powerful, I can change my mind, if I say for example “let’s attack… let’s have a revolt, a coup d’état” and I have tens of millions of people listening to me who think “you are right, let’s do this coup d’état”, in the end it’s over!

**Table 6 ijerph-20-07013-t006:** Strategies of intervention.

Themes	Fragments of Discourses
Focus on the individual	
(e.1) Psychological help and treatment center	They go to the psychologist. That should help. Also, because it is an addiction like any other, even an addiction like drugs! The Government must do something like it does in Switzerland for drug addicts… there are houses, places where they are hosted, to be debilitated… So, they should create these Centres to learn to use the internet well, to be detoxified from the internet. Yes. Create this kind of place, where people who can go and get help… The psychologist could help because he or she can give you advice. Look you’re not crazy, you know? You don’t necessarily go there if you have a problem… That is, you go to the psychologist when you also have an addiction problem. Anyway, he or she gives you advice and tells you how to resolve this situation…
Focus on the interpersonal environment	
(e.2) Parental monitoring on children’s internet use	Parents should put time limits on apps. If I were the parent of a child who spends a lot of time on the PlayStation, I’d tell them not to play for long… if they play for an hour, then I’d tell them “Turn off it” and they don’t cooperate and keep playing… I’d remove the PlayStation! My mother would tell me “Stay an hour” if I stayed longer than that hour the Wi-Fi button went on and off. I would remove WhatsApp, Instagram, Facebook for children and adolescents.
(e.3) Appropriate parental educational style and family climate	It depends on the parents. From how the children are raised. Yes, but also the fact of taking the child to the park, going out where there are other children… In my opinion, parents could encourage kids to have hobbies, I grew up with the bike, I was always cycling, I continue even now and I use the smartphone very little when I am at home because I do sports, when I’m not playing sport I study, or I go out, I am at school, so I use the smartphone just to get organised or to know if I have to do something, but I really do have a limited use and having hobbies or doing recreational activities would make the child disconnect from the phone. In my opinion we should try to plug the time gaps, close these empty periods… The family always teaches to do homework first, do important things first. First what you “must” do. Then if you have time in the evening, if you are not tired, use your smartphone… but to watch a video or to play a little, and not to stay connected for hours and hours. Parents must be able to say not to follow the crowd, the child must be himself, I have transmitted these values… that is, it is an important thing for us. You don’t have to follow the crowd. Parents have to be capable too… In my opinion, the only thing that makes you happy is your family (and not the followers you have). Maybe I prefer to have a nice close family… Talk to me instead of talking to your smartphone!”, there must also be responsibility on the part of the parents…
(e.4) Friends’ support	Friends help you go out at night! They tell you “What happened? Why aren’t you going out anymore?”. Friends notice the change, they see you when you no longer have the same habits, you don’t go out anymore… they immediately ask you “Are you going out tonight? Why aren’t you going out? “, I say this because I do the same the group of friends notices it… but there are few of them though. If it’s a very close friend, he really comes to your house and helps you. If you don’t want to go out, he just pushes you out. If he has a problem, he tells you “Let’s go out together, talk, face it together rather than stay at home and take refuge in social media…”. Friends. Maybe if they see that you are always online, they could help you heal. [In reference to the possibility of talking to a friend in times of difficulty] But if the friend is OK, he is not always on your side. I mean, if you do something and you confide in your friend, he tells you “This is wrong”. If you still have a friend and you also know that you can count on them in difficult times… you confide in them and not on the internet.
(e.5) Education activities to promote informed use of the internet	Discuss these issues for up to one hour a week, in a regular group appointment. It must be compulsory, otherwise you won’t change, because the problem isn’t recognised… that’s why the help desk is no use. Current issues are interesting. Show people how to use the internet. Even at school… there is not much information… Learn the basic assumptions of the web!!! There is a threshold… when you cross that threshold, there are consequences. Then you consider those consequences, to realise that you are wrong and if you are capable of going back For example, the school could very well make a program, I don’t know, an App with all the books downloaded into your device, this would also favour the most suitable use of a child’s smartphone and maybe this could even decrease the incidence of being online via smartphone perhaps playing games or watching videos.
Focus on social policies	
(e.6) Opportunities for socialisation and leisure	I would do outdoor camps, where they do manual projects as a community. In contact with nature. For example, in the park and you create a typical art day or where you are busy doing something, your interest might be aroused more. But, also, simply even the library… maybe they could organise special days… you are also more encouraged to go there and maybe start reading a book without using the smartphone. I would suggest doing more physical activity, attending sports centres where friendships are consolidated, and bonds are established. I think anyone of this age [adolescence] must have a hobby. In my opinion, those with a hobby don’t need to spend a lot of time on the internet. The method is to have something to do, something to take your mind off things. For example, if there are some guys who are on the internet in the afternoon and don’t have an interest, something that prevents them from always going on the internet… a sport, if you play football, in the afternoon you have to go and play rather than stay on the internet…
(e.7) Policies to limit access and ban inappropriate sites	In my opinion, removing the self-harming pages… Restrict certain games and forbid them for minors, or even them to visit sites. I think sites must be banned and so when you are a minor, you cannot enter, while for adults you must give consent, that is, write that you are already 18 years old and therefore you can enter the site or the game. Putting your e-mail address and password, so you are traceable, then one does not enter. Only the e-mail and password, however, no other information. In the sense that inappropriate sites, that are free, that everyone can see, should be deleted…
(e.8) Promotion of more healthy identification models	For me, ridding society of the idea that if one person prevails over the other, he/she somehow feels cool or more important, and instead establishing as a model that what makes the person feel cool and great is helping those in difficulty, as an example to imitate.

**Table 7 ijerph-20-07013-t007:** Opportunities and limits of the digital worlds.

Themes	Fragments of Discourses
Opportunities	
(f.1) Performing daily tasks more easily and/or learning more quickly	For example, I play the guitar and I study singing and, in any case, I need the smartphone because I have the chords online, to study singing I must necessarily find the soundtrack on the internet. When you ride a bike, don’t you use your smartphone to see how far you have travelled? I meant that I learned Spanish, I never studied it at school, I studied French in middle school, but by watching TV series [online] with Spanish subtitles I was able to learn and now I speak it well.
(f.2) Spending time	[…] I mean, there are now online books that in my opinion don’t have the same thing as the paper book. The smell of the pages, the sound of leafing through, the things to imagine. They don’t lead you to the imagination that a real book brings you. So, then you confuse the virtual thing with the real thing. The real thing is the book and the virtual thing is the digital book. Yes, but it’s also an advantage because you don’t have to carry a book or a lot of them. I mean for schoolbooks, not for those to read. Then many of us watch TV series on the smartphone, after studying, in the evening, instead of reading. In my opinion it is seen as a way to relax, in the evening after I finish studying, I can’t wait to go to bed, stay on the smartphone to relax mentally because just reading a book, or not even television relaxes me so much. We hardly watch the television anymore, it is in the room, but you stay on your smartphone anyway, you watch the TV series on the smartphone, at the most you use the PC, then you turn it off and go to sleep… On the smartphone, you can choose what you want to see, on television you are limited, then there are no advertisements, whatever you do immediately, requires no mental effort.
(f.3) Information seeking	But we also ask for information or rely on the internet rather than asking someone in person. If you have to look up a word in the dictionary, everyone is looking for it on the internet anyway. This is a benefit! Yes, this can be a benefit… If you have to do a Latin translation it is very useful, how could you do it without the internet, it is impossible… But a person faced with the choice between a dictionary and the internet, will obviously choose the smartphone because it is faster.
(f.4) Sharing idea or interests and approaching people	On social media, when you have a certain group, you can express your opinion on a certain topic and thus you feel like you are participating in something. In reality, you may find a person to share with, but sometimes not. So, you don’t feel totally included in a reasoning; instead on the internet you can find others agreeing on. Thanks to the internet, more boys and girls are falling in love… Because they meet each other online.
Limits	
(f.5) Meeting people who are not what they seem	It makes no sense to talk about many things through messages and then when you are live it is as if I do not know you. On the internet, the approach to a girl is better, you can be cool, then you meet her live and you can’t say a word… you are petrified… In my opinion, the relationship that is established with a person is different, because maybe on Instagram you can joke, then you see the person and maybe you can’t even say hello. On social media, from the photo it may be a fifteen-year-old boy, then in reality there is a forty-year-old.
(f.6) Being exposed to untrustworthy/dangerous people or applications	For example, anyone could write to you on Instagram, maybe paedophiles… I message with somebody and I think they are one person and then they turn out to be completely different, even in terms of photos not everyone is the same, they can put a picture of anyone and pretend… Those are dangers! Even right now on the sites, that is very often, you go to look for that person on Google or on Instagram, as you click so many pages come out that say “If you are 18 years old click Accept”. Even dating sites are dangerous. If a girl with serious problems googles the word “self-harm”, a message appears which says: “The contents are potentially dangerous, they can be etc…” and “Accept or Cancel”. If she has problems and wants to see it, she can click “Accept”. In the end that’s what happens, it is not that it stops you, if you want to do it either there or on Google… or in any case it is written “Prohibited for minors under 18 years, if you are 18 years old click here” and you even if you are not 18 you click it. […] in other words, everyone does what they want, even if it will hurt them.
(f.7) Being exposed to untrustworthy news	There is also the problem that we no longer watch the news, now we are informed via the smartphone, and maybe the news is not as reliable as the TV news, because okay they can say what they like but they have more visual evidence compared to news on the phone. If you go to look for anything on the internet you can get twenty-seven different results, there are different opinions from other people who may not even understand but want to have their say…
(f.8) Being exposed to a false view of life	[Influencers] give you a false idea of life, as my mother says. My mother hates CF, because she says that she shows you life as it is if you have money, if you are rich, if you are famous, but in reality, life is not like that! In the end, they [influencers] only show good things in life, maybe they have problems too, because everyone has problems, poor, rich, with money or not, famous… only that by giving this false vision of life you think “If I become famous, I’ll do all these things, then I’ll be happy”, but in reality, it is not like that…

**Table 8 ijerph-20-07013-t008:** Needs that adolescents try to satisfy by surfing the net.

Themes	Fragments of Discourses
(8.1) People you would not know how to approach in the offline world	We use the internet for things we don’t know how to do in reality. For example, relating to people! We use the smartphone to have a little more courage to relate to someone.
(8.2) Find what you do not find in reality	I use the internet to have a dialogue. Knowing that there is someone who shares what you say. That you are not alone on a certain matter. Feeling part of something. But also expressing yourself or finding what you don’t find in reality.
(8.3) Nourish self-esteem and self-image	It is also a way to feel good about yourself… if a person doesn’t like themselves, I know by taking a photo, they can see themselves with different eyes. Even through the online feedback. In fact, someone who has low self-esteem and then putting some photos they get some nice comments you also have an increase in self-esteem. Satisfaction. On Instagram I have more followers than you and therefore I am superior… You buy them, people do this too. They buy followers! On the other hand, I am fixated on Likes. As soon as somebody stops following you, I immediately check who they are so as to stop following them. Presenting themselves as what they wanted to be in life… and what they failed to be. In reality you constantly feel judged, not like behind a smartphone which is easier for you. The internet allows you to create your own little world, you can customize what you like and you can be whoever you want within the internet.

## Data Availability

The data presented in this study are available on request from the corresponding author.

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
