# Peer review of "What Adolescents Have to Say about Problematic Internet Use: A Qualitative Study Based on Focus Groups"

_ijerph, 2023, doi:10.3390/ijerph20217013_

Round 1

Reviewer 1 Report (Previous Reviewer 2)

Comments and Suggestions for Authors

The paper is a revise version that showed significant improvement in describing the methodology of this qualitative study about problematic internet use. The theoretical part, discussion and conclusions has been extended and improved and the paper is now good for publication.

Reviewer 2 Report (Previous Reviewer 3)

Comments and Suggestions for Authors

The paper is now sufficient for publication.

This manuscript is a resubmission of an earlier submission. The following is a list of the peer review reports and author responses from that submission.

Round 1

Reviewer 1 Report

Comments and Suggestions for Authors

I suggest to organize this manuscript a bit better. The result section starts with good explanation of seven macro-strategies. This sets a stage for what to expect. However, when I keep reading further, it's tough to follow under which macro strategy explanation is provided. It would be better if the text in manuscript is appropriately given bullets or numbers, so readers can identify which macro-strategy is being followed and sub-sections are part of which macro-strategies. 

Author Response

REVIEWER 1

I suggest to organize this manuscript a bit better. The result section starts with good explanation of seven macro-strategies. This sets a stage for what to expect. However, when I keep reading further, it's tough to follow under which macro strategy explanation is provided. It would be better if the text in manuscript is appropriately given bullets or numbers, so readers can identify which macro-strategy is being followed and sub-sections are part of which macro-strategies. 

AUTHORS’ ANSWER: In the submitted format, unfortunately, the indentation of some lines or the font size of the labels given to the categories changed and this probably made reading the results even more complicated. We apologise for this. We have corrected these typos in the layout and - as suggested by the reviewer - numbered the seven macro-categories and sub-sections to better display the contents identified. Furthermore, a brief overview for each of the macro-categories has been inserted to guide the reading.

Reviewer 2 Report

Comments and Suggestions for Authors

Thank you for the opportunity to review this paper.

The paper presents the results of a thematic analysis on focus-group discussions about problematic internet use in adolescents.

‘’All procedures were approved by the Ethics Commission for Research in Psychology of the Department of (NAME OBSCURED TO PRESERVE ANONYMITY)  (25 March 2021; protocol no. 0056300).’’

I don’t think there is justification to not give the name of the department, the authors institutional affiliation and name are visible so can you justify why you obscure the name.

Did

Regarding methodology can you describe in more details how the thematic analysis was carried out.

Did you use any software for the analysis?

You mention that some difference between researchers emerge. Did you calculate an inter-rater agreement?

You mentioned that three macro topics were proposed for debate within each focus group:

‘’(1) when can we talk about PIU (i.e., what – in their view – constitutes a problematic use? How is it recognizable?); (2) how can PIU be explained? (i.e., what are – in their view – the causes  of a problematic Use?);

(3) how can PIU be prevented? (i.e., what are – in their view – the  target and the strategies of intervention?).’’

Did you use an interview guide or just these 3 questions?

You mentioned that ‘’TA allowed us to identify seven macro-categories. Five relate to PIU: Definition; Symptomatology; Impact; Determinants; Intervention strategy. Two further topics focus  on the common condition of living in a digital world: opportunities and limits of the digital world and needs adolescents try to satisfy by surfing the net and which the offline world does not fulfil. For each macro-category, specific themes and fragments of discourses are reported below.’’

The paper would gain in clarity if this definition as emphasized and then followed by the examples in the discussions in the focus group.

If you have coded all the discussions in the focus-group in order to identify this topics can you present also some quantitative information about the frequency of these topics.

Author Response

REVIEWER 2

Thank you for the opportunity to review this paper.

The paper presents the results of a thematic analysis on focus-group discussions about problematic internet use in adolescents.

REVIEWER’S COMMENT: ‘’All procedures were approved by the Ethics Commission for Research in Psychology of the Department of (NAME OBSCURED TO PRESERVE ANONYMITY)  (25 March 2021; protocol no. 0056300).’’ I don’t think there is justification to not give the name of the department, the authors institutional affiliation and name are visible so can you justify why you obscure the name. Did

AUTHORS’ ANSWER: The name of the department and of the university has been inserted:

All procedures were approved by the Ethics Commission for Research in Psychology of the Department of Human and Social Sciences of the University of Salento (Lecce, Italy) (25 March 2021; protocol no. 0056300).

***

REVIEWER’S COMMENT: ‘Regarding methodology can you describe in more details how the thematic analysis was carried out. Did you use any software for the analysis?  You mention that some difference between researchers emerge. Did you calculate an inter-rater agreement?

AUTHORS’ ANSWER: we did not use of any software for the analysis. In the revised manuscript, the procedure followed has been explained in more details:

“We began the analytical work by listing the themes the interviewees talked about within each of the questions proposed and labelling them: for example, for the topic “when we can talk about PIU”, sentences focusing on the time spent on the internet were grouped under the label “high frequency of internet use”, sentences focusing on the compulsive desire to get connected were collected under the label “craving” and so on. We then grouped themes into macro-categories (e.g., “high frequency of internet use”, “craving”, “interpersonal conflicts” were grouped under the macro-categories “symptomatology”).

TA was performed by two groups of three researchers (a totally of six research collaborators) that worked independently. Each theme was consolidated after an intra-group (three-researchers) discussion and subsequently through an inter-group (six-researchers) discussion. Thanks to these comparisons, common judgments and differences among researchers emerged in the formulation of a synthetic label. The common judgment led to the selection of the theme identified; any divergence was resolved by referring to the scientific literature (e.g., labels referring to the compulsive desire to get connected were redefined in terms of “craving”), or to the more general discussion, thus seeing how theme X is connected within a sequence of questions and answers (e.g. the label “social detachment” was used to group both utterances referring to symptoms and utterances referring to the impact of PIUs, because participants referred to social detachment both in the attempt to define the symptoms of PIU and its consequences). When the disagreement could not be resolved between the researchers, an external researcher (research coordinator) was involved as supervisor.”

We did not calculate an inter-rater agreement. In the most cases, researchers proposed labels which were convergent in their meaning and their work was only to identify the label that could best describe this meaning.

***

REVIEWER’S COMMENT: You mentioned that three macro topics were proposed for debate within each focus group: ‘’(1) when can we talk about PIU (i.e., what – in their view – constitutes a problematic use? How is it recognizable?); (2) how can PIU be explained? (i.e., what are – in their view – the causes  of a problematic Use?); (3) how can PIU be prevented? (i.e., what are – in their view – the  target and the strategies of intervention?).’’ Did you use an interview guide or just these 3 questions?

AUTHORS’ ANSWER: As specified in the revised manuscript:

“An interview guide was defined as a list of questions [53], which directed conversation within each focus group towards three research topics: (1) when can we talk about PIU (i.e., what – in their view – constitutes a problematic use? How is it recognizable?); (2) how can PIU be explained? (i.e., what are – in their view – the causes of a problematic Use?); (3) how can PIU be prevented? (i.e., what are – in their view – the target and the strategies of intervention?). Participants were encouraged to say whatever came to mind in response to these topics and responding in the manner that is deemed most appropriate, taking into account that the objective of the investigation was to collect their subjective view of PU.  The focus group leaders took care to foster an open conversation, allowing each participant to explore each of the three topics in the way useful and meaningful to them and also allowing divergent points of view to be expressed.

***

REVIEWER’S COMMENT: You mentioned that ‘’TA allowed us to identify seven macro-categories. Five relate to PIU: Definition; Symptomatology; Impact; Determinants; Intervention strategy. Two further topics focus  on the common condition of living in a digital world: opportunities and limits of the digital world and needs adolescents try to satisfy by surfing the net and which the offline world does not fulfil. For each macro-category, specific themes and fragments of discourses are reported below.’’

The paper would gain in clarity if this definition as emphasized and then followed by the examples in the discussions in the focus group.

If you have coded all the discussions in the focus-group in order to identify this topics can you present also some quantitative information about the frequency of these topics.

AUTHORS’ ANSWER: In the submitted format, unfortunately, the indentation of some lines or the font size of the labels given to the categories changed and this probably made reading the results even more complicated. We apologise for this.We have corrected these typos in the layout and numbered the seven macro-categories and sub-sections to better display the contents identified.

An analysis of the frequency of the categories seemed impractical in view of the nature of the texts collected. As specified in the discussions:

“(…) the discourses of adolescents adopted a connectionist, rather than disjunctive, logic in approaching the topics proposed: the statement of one was commented on, deepened or supplemented by another, so that the overall discourse emerging from the focus groups held together viewpoints, subjective ways of feeling, direct and indirect experiences. For this reason, when analysing the transcripts qualitatively, we chose to identify and code the proposed themes, but it did not seem possible or appropriate to analyse them in terms of frequencies.”

Reviewer 3 Report

Comments and Suggestions for Authors

Thank you for the opportunity to read this manuscript on the important topic of problematic internet use. Although this is a timely and interesting topic the methodology (thematic analysis of student interviews) lacks scientific merit. There are additional concerns I have with the quality and clarity of the writing throughout this paper. For these reasons I do not believe this manuscript should be accepted for publication in this journal.

Page 2 line 50: There are eight lines of text in this single sentence. The message comes across as highly jumbled and unclear to the reader.

Page 2 line 58: Although this example is interesting, it could be summarized in a much more concise manner. Also, some scientific literature should be used to support what is implied by this example.

Page 2 line 88: This could be summarized more concisely as “in-line with the ecological systems perspective, adolescents (the most at-risk demographic for PIU) develop within an intricate and multi-layered social dynamic.

Page 3 line 118: Useful approach but I do not think the stage is effectively set in the intro. It is convoluted by run-on sentences and the differing perspectives should be presented more concisely and clearly.

Page 6: The results are highly descriptive and no inferential statistics were conducted. Although they offer some interesting insight on student perceptions of PIUs they are not scientific and could instead be published in a blog or an addiction-specific journal (international journal of addictions).

Page 8: The insight implied by the student comments merely show that students have an adequate intelligence pertaining to PIUs. However, they do not really offer anything new to this topic and no policy implications can be gleaned that have not already been proposed elsewhere.

Comments on the Quality of English Language

Quality is acceptable but the writing is neither clear nor concise.

Author Response

REVIEWER 3

Thank you for the opportunity to read this manuscript on the important topic of problematic internet use. Although this is a timely and interesting topic the methodology (thematic analysis of student interviews) lacks scientific merit. There are additional concerns I have with the quality and clarity of the writing throughout this paper. For these reasons I do not believe this manuscript should be accepted for publication in this journal.

AUTHORS’ANSWER: We have tried to improve the clarity of the writing and to better highlight why, from our theoretical standpoint, qualitative study can be important for guiding intervention strategies and what implications for policy can be considered in the light of our findings.

***

REVIEWER’S COMMENT: Page 2 line 50: There are eight lines of text in this single sentence. The message comes across as highly jumbled and unclear to the reader.

AUTHORS’ANSWER: The sentence has been revised and synthesised. Emphasis was placed on the concept that the view of PIU in terms of disease is a way of constructing an explanation of a problem, not a description of a fact.

***

Page 2 line 58: Although this example is interesting, it could be summarized in a much more concise manner. Also, some scientific literature should be used to support what is implied by this example.

AUTHORS’ ANSWER: The example has been summarized in just a few lines. it serves only the purpose to highlight how different models of explanations and condition of observation end up defining different problems.

“In the well-known rat park experiment, Alexander and colleagues [30] placed several rats in a cage and provided them with colourful balls, good rodent food, tunnels in which to exercise; in such an environment, the rats seemed uninterested in taking cocaine-contaminated water, unlike rats placed in a cage without other rats to play with or any other stimulus.”

Scientific literature supporting the idea that the quality of the interpersonal and social environment plays a central role in the developing and maintaining of PIU or other forms of psychological distress have been cited at page 3.

“Several studies support the idea that the interpersonal and social-cultural environment is critical in understanding PIU as well as other addictive behaviours. For example, low family support, family functioning and parental monitoring, school climate and deviant peer affiliation have been identified as leading factors in developing maladaptation problems and in the search for dubious sources of gratification, including video-games [32-34]. Other studies support the importance to better investigate dimensions related to the needs and demands addressed to the social environment: low social capital (e.g. in-personal and social trust, size of social networks, social support), feelings of anomia, feelings of loneliness and alienation were found to have a significant association with PIU [35, 37], gambling [38, 39], as well as other indicators of psychological distress [40-44]. Meanwhile, the prevention programmes are seen to have overlooked the need to address psychosocial problems, including areas such as supporting the development of healthy identity, preventing loneliness, improving opportunities for socialization and leisure time [31, 45].”

***

REVIEWER’S COMMENT: Page 2 line 88: This could be summarized more concisely as “in-line with the ecological systems perspective, adolescents (the most at-risk demographic for PIU) develop within an intricate and multi-layered social dynamic.

AUTHORS’ANSWER. The sentence has been summarized as follows:

“(…) if we think of adolescents – the most at-risk demographic for PIU – they are also children, students, citizens, namely they develop within an intricate and multi-layered social dynamic and specific family environments, specific school environments, specific neighbourhoods which can offer resources or place constraints on their adaptive potential.”

***

REVIEWER’S COMMENT: Page 3 line 118: Useful approach but I do not think the stage is effectively set in the intro. It is convoluted by run-on sentences and the differing perspectives should be presented more concisely and clearly.

AUTHORS’ANSWER. The introduction has been revised, in an attempt to better finalize the discourse. Arguing in three points: a) the PIU view in terms of disease is a way of telling and explaining a problem, b) like any other model, this way carries with it important implications. We emphasized two of them:

First, the scant attention given to the role played by the interpersonal and social environment

“Consistently with this approach, prevention strategies are found to focus on vulnerable/at-risk individuals or age groups and their cognitive, emotional and behavioural components: develop conscious Internet use and effective use of time, improve communication skills, increase self-esteem and reduce anxiety, provide knowledge on different types of addiction, common symptoms of dependency, with short- and long-term impacts being among the main areas addressed by the intervention (for a review: [31]). However, if we think of adolescents – the most at-risk demographic for PIU – they are also children, students, citizens, namely they develop within an intricate and multi-layered social dynamic and specific family environments, specific school environments, specific neighbourhoods which can offer resources or place constraints on their adaptive potential. Several studies support the idea that the interpersonal and social-cultural environment is critical in understanding PIU as well as other addictive behaviours. For example, low family support, family functioning and parental monitoring, school climate and deviant peer affiliation have been identified as leading factors in developing maladaptation problems and in the search for dubious sources of gratification, including video-games [32-34]. Other studies support the importance to better investigate dimensions related to the needs and demands addressed to the social environment: low social capital (e.g. in-personal and social trust, size of social networks, social support), feelings of anomia, feelings of loneliness and alienation were found to have a significant association with PIU [35, 37], gambling [38, 39], as well as other indicators of psychological distress [40-44]. Meanwhile, the prevention programmes are seen to have overlooked the need to address psychosocial problems, including areas such as supporting the development of healthy identity, preventing loneliness, improving opportunities for socialization and leisure time [31, 45].”

Second, the scant attention given to the point of view of the potential problem bearers, their interpretative criteria, their subjective experience. The study sets out to analyse their point of view of adolescents and their experience of using the internet because, within the socio-constructivist perspective guiding the work, this point of view organises not only ways of acting on the online world but also ways of relating to the contents that are proposed to them to preventing and contrasting PIUs

“(…) the constructivist approach invites us to conceive and value the sufferer or the individual at risk as a semiotic subject [48] whose ways of interpreting experience also organize the ways they live their life, make sense of their problem, and relate to the health services [47, 49]. Recognising the regulatory value of meaning on people ways of living and acting has important implications for research and intervention. It suggests the opportunity to give more space to qualitative research in order to carry out an in-depth exploration of people’s ways of thinking and relating to risks. Qualitative research suggests, for instance, that the youths’ criteria may therefore deviate greatly from that of the “expert” risk assessors or adults [50, 51, 52]. Ignoring this aspect exposes to failure any prevention strategy that takes it for granted that there is agreement on what is considered problematic. “

***

REVIEWER’S COMMENT:

Page 6: The results are highly descriptive and no inferential statistics were conducted. Although they offer some interesting insight on student perceptions of PIUs they are not scientific and could instead be published in a blog or an addiction-specific journal (international journal of addictions).

: Page 8: The insight implied by the student comments merely show that students have an adequate intelligence pertaining to PIUs. However, they do not really offer anything new to this topic and no policy implications can be gleaned that have not already been proposed elsewhere.

AUTHORS’ANSWER.

We can understand that within certain approaches to research, the use of quantitative data and inferential statistics is considered a sine qua non for scientific research. At the same time, it can be recognised that there is a qualitative tradition that answers other questions and purposes. This is clearly not the place for a debate on this topic but let us try here to clarify our point of view and how we have revised the text to better highlight outcomes of the study and implications for preventive strategies. You comment, although very critical, served us to visualise that these aspects needed to be made clearer.

The purpose of the study was to explore the point of view of adolescents, a privileged target of intervention strategies but, paradoxically, little explored with respect to the themes addressed in this article. What do they think about PIUs, how do they explain them, how - from their point of view - could they be countered?

As we have tried to argue, their point of view is crucial (though certainly not exhaustive) for understanding what to set intervention strategies on.

It was not our objective to understand whether or not they demonstrate intelligence with respect to PIUs, although a) their awareness is not taken for granted (often scholars, families, teachers tend to think of them as unaware group, having poor risk-judging skills) and b) noting their awareness of the risks associated with PIUs seems important because it suggests that knowing the risks is not necessarily sufficient to avoid problematic engagement (we also know this with respect to other addictions: most smokers are aware of the health risks they face but this knowledge is not sufficient to stop smoking).

For the frame that organizes the present study, the value of the collected narratives is given not by their mere ability to represent “reality” or to explain the “nature” of Problematic Internet Use - much research based on quantitative data and inferential statistics can be used for this purpose. Rather, their value lies in the fact that their perspective organizes ways to relate to the internet, as well as to evaluate the intervention addressed to them. Furthermore, their perspective offer insight on how they represent the relationship between internet pattern of use and what happened in their life and social environment.

Our study aimed to explore whether adolescents recognize PIU, how they explain PIU (its determinants), and whichstrategies for intervention they suggest. As we emphasized at the beginning of discussion:

“Touching on these issues, the adolescents participating in the focus groups also recounted opportunities offered by the use of the Internet and questions of identity and recognition that they try to solve through social networks, even when they cannot describe this use as problematic. In other words, they tell us the meaning of their involvement and signal the importance of understanding it; an approach to Internet use that is only marginally present in the literature [5, 60, 61]. It is also worth noting that in giving meaning to their internet use and in citing both causes and solutions, participants emphasize the role played by the social environment. In this perspective, adolescents are soliciting us to adopt understanding models and intervention strategies that consider not only aspects related to individual vulnerability but also the quality of interpersonal relationships (e.g., educational styles, behavioural patterns proposed by adults) and their living environments (e.g., socialisation opportunities and leisure activities).”

The thematic analysis of the focus groups reveals three different ways of understanding the “problem”, focused on the “addictive character” of the medium and/or of its applications, on the individual, their relational network, or the socio-cultural context they belong to. On a first reading level, we could therefore conclude that the adolescents interviewed do not add anything to what quantitative research has already shown. However, as we emphasized in the discussion:

“(…) it is worth noting that the discourses of adolescents – differently from what is usually the case among scholars – adopt a connectionist, rather than disjunctive, logic in identifying determinants of PIU. In the focus groups, there was no array of opinions; the statement of one was commented on, deepened or supplemented by another in the search for an explanatory framework that held together viewpoints, subjective ways of feeling, direct and indirect experiences. In the overall picture emerging from the narratives, PIU appeared to be the outcome of a psychological dynamic emerging from the interaction of individual, interpersonal and sociocultural dimensions. PIU does not belong to adolescents – the participants seemed to be saying – not only because a problematic approach to the internet is recognizable even in the adult reference-points (“Parents set a bad example when they don't pay any attention to us for being on the smartphone!”), but also because the interpersonal environment plays an important part, in seeing or not seeing individual or age vulnerability, in whether or not support is provided for the difficulties of growth or, more broadly, of life, in whether or not healthy models of identification are offered, in whether limits and rules are set, and whether or not alternative ways of spending time, having fun and socializing are proposed”.

We think that significant implications for policy can be recognized. In the attempt to show the heuristic and pragmatic potentiality to take into account and value the views of adolescents, in the revised manuscript we have tried to develop implications for policy.

The Conclusion section has been revised as follows:

“Many studies on PIU have focused on adolescents, recognised as the group most at risk, to whom prevention interventions should be directed. Nevertheless, little research has been conducted to understand the perspectives of the so-called “digital natives”.

The current study tried to bridge this gap, in the idea that no preventive strategy can be effective without an understanding of the interpretative criteria that adolescents use to give meaning to the use, even problematic, of the Internet. Just think how empty, meaningless a discussion about the risks of the Internet might seem to adolescents who are aware of the impact that use can have on their psycho-physical well-being, their relationships, their performance at school. Against a widespread view of adolescents as being unaware and with poor risk-judging skills, the adolescents we interviewed seem aware of the risks that can occur in surfing the internet and of the negative outcome that an excessive use of the internet can have on their life. At the same time, they suggest that the knowledge of these risks is not always enough to prevent PIU because issues related to self-image, perceived quality of friendship and family relationships, lack of alternative and meaningful activities, models and criteria for success proposed in the broader social environment are at stake. The findings thus tell us something not only about how adolescents understand PIU but also how they understand themselves in relation to their significant others and in relation to culture in the broadest sense [82].

Significant implications for policy can be recognized if this perspective is taken into consideration. First, any intervention that is limited to the specific domain of PIU is likely to have limited efficacy, given that adolescents (and more generally, people) shape their way of using internet not only according to internet domain‐specific beliefs and expectations, but according to their ways of representing themselves and their social identity within specific relational contexts [83-85].Preventing PIU is therefore not simply a matter of controlling access to the Internet, but of offering spaces for listening and reflection on the ways in which adolescents try to respond to their own needs for recognition and sociality. Second, if these ways are encouraged by the social environment, strategies should be sensitive to how adolescents’ network of interdependencies (e.g., family, friends, teachers) may frame and influence the ways they think and act, and include this network in the range of action. More broadly, preventing PIU means rethinking – sometimes radically – material, relational and symbolic resources (e.g., socialization channels, relationship models, educational styles, modes of communication, criteria of social recognition) that the environment makes available to interpret their own experience, face problems, and make the future thinkable.”

A further sub-section was inserted to highlight limitations and future direction of research, also in the perspective to recognize how the integration of a qualitative and quantitative approach can offer greater insights.

“Some methodological limitations of our study should be considered. First, since it is based on a convenience sample, the results cannot be generalised and have to be related to the specific cultural context under analysis. Second, the qualitative analysis of how adolescents represent and explain PIU could be improved by considering quantitative measures accounting for their Internet usage patterns. As a matter of fact, we do not know how the participants in our research characterize themselves in this respect, and we cannot therefore exclude the possibility that the discourses collected refer to adolescents who make a balanced and adaptive use of the Internet. Future research should consider other factors such as psychological well-being, parental monitoring, perceived social support, sense of belonging to the community, to look more closely at the way individuals, their system of activity and the socio-cultural scenario interact with each other in constructing the ways adolescents represent and use the internet.”

Round 2

Reviewer 3 Report

Comments and Suggestions for Authors

The authors have taken considerable steps to improving the merit of this manuscript. I still believe it is a stretch to conclude that "these results can provide valuable knowledge towards the guidance of intervention approaches related to PIUs". Particularly in qualitative studies, it is vital to consider the SES, culture, and educational background of the participants. The authors have now acknowledged this limitation. This study can be published in its current form.

Comments on the Quality of English Language

NA

Author Response

Comment of the reviewer 3

The authors have taken considerable steps to improving the merit of this manuscript. I still believe it is a stretch to conclude that "these results can provide valuable knowledge towards the guidance of intervention approaches related to PIUs". Particularly in qualitative studies, it is vital to consider the SES, culture, and educational background of the participants. The authors have now acknowledged this limitation. This study can be published in its current form.

AUTHORS’ ANSWER

Thank you for your comment. As suggested, the following sentence has been added in the ‘limitations’ section:

“First, since it is based on a convenience sample, the results cannot be generalised. Characteristics such as the socio-economic status, educational background and cultural background of the participants can play an important role with respect to how and what teenagers have to say about PIUs”.
